# Comparative Otolith Morphology of Two Morphs of *Schizopygopsis thermalis* Herzenstein 1891 (Pisces, Cyprinidae) in a Headwater Lake on the Qinghai-Tibet Plateau

Jialing Qiao [1,2], Ren Zhu [1], Kang Chen [3], Dong Zhang [2], Yunzhi Yan [2] and Dekui He [1,*]

1   Institute of Hydrobiology, Chinese Academy of Sciences, Wuhan 430072, China; qiaojialing9406@126.com (J.Q.); zhuren@ihb.ac.cn (R.Z.)
2   Collaborative Innovation Center of Recovery and Reconstruction of Degraded Ecosystem in Wanjiang Basin Co-Founded by Anhui Province and Ministry of Education, School of Ecology and Environment, Anhui Normal University, Wuhu 241002, China; kentmars@foxmail.com (D.Z.); yanyunzhi@ahnu.edu.cn (Y.Y.)
3   Yangtze River Fisheries Research Institute, Chinese Academy of Fishery Sciences, Wuhan 430223, China; chenkang992@163.com
*   Correspondence: hedekui@ihb.ac.cn

**Abstract:** Teleost otoliths provide a pivotal medium for studying changes in population structure and population dynamics of fish. Understanding the otolith-fish size relationship and intraspecies variation in otolith morphology is essential for the accurate assessment and management of fishery resources. In our study, we aimed to estimate the relationships between otolith morphological measurements and fish length, and detect differences in the otolith morphology of planktivorous and benthivorous morphs of *Schizopygopsis thermalis* in Lake Amdo Tsonak Co on the Qinghai-Tibet Plateau (QTP). Both morphs exhibited strong linear otolith-fish size relationships; otolith morphology was sexually dimorphic in each morph; the morphs differed significantly in otolith shape and size (e.g., posterior side, the region between the posterior and ventral otolith, otolith length, circularity, and surface density). In addition, we found that the differences in otolith morphology between morphs are related to habitat preferences, diet, and growth. Basic data on the biology of *S. thermalis* are essential for poorly studied Lake Amdo Tsonak Co, and our study emphasizes that intraspecific variation in otolith morphology should be taken into consideration when differentiating stocks, populations, and age classes based on otolith morphology.

**Keywords:** intraspecific variation; otolith shape; otolith-fish size relationship; plateau lake; *S. thermalis*





## 1. Introduction

Teleost otoliths provide a pivotal medium for studying many aspects of fish biology, including bioacoustics, systematics, and ecology, especially for fishery stock assessments [1–4]. Otoliths can provide vital information about fisheries at different scales. This information includes features of individuals, such as growth, reproduction characteristics, and migration pathways; the spatiotemporal structure of fish populations and stocks as affected by the recruitment process, mortality, and anadromy, and the historical environment of the ecosystem [5–7].

Otoliths are acellular solid calcium carbonate biomineralization in the inner ears of fish that are characterized by continuous deposition [8] and metabolic inertia [9]. Thus, they are not only good timers (e.g., age, growth, and life history information) but also a mechanoreceptor for processing acoustic (auditory sense) and postural (movement balance) information [1,3]. The morphology of otoliths (i.e., otolith shape, size, and mass) is an important tool in fish biology. Otolith morphology is usually associated with movement, auditory, and other sensory functions in fish [3,10]. For instance, Schulz-Mirbach et al. [3] reviewed that larger otolith provides improved auditory sensitivities in *Ophidion rochei* [11]

and zebrafish larvae [12], and a better sense of balance in *Carapus acus* [1,11]. Compared to fish that possess spherical or ellipsoidal otoliths and feature simple movements, fish with otoliths showing an increase in shape complexity may have higher mobility [13].

In recent years, otolith morphology has been shown to exhibit high interspecific variation but a relatively less intraspecific variation, which makes it frequently utilized in population and stock identification [4,14,15]. Although less variation in otolith morphology generally occurs within species than between species, intraspecies variation in otolith morphology is usually used to discriminate sexes, stocks, populations, and age classes in many fish species, such as *Lutjanus campechanus* [16], *Coilia nasus* [17], and *Porichthys notatus* [18]. In addition, otolith morphology is also recommended in cases of taxonomic confusion caused by high levels of intraspecific morphological plasticity and low levels of intraspecific genetic differentiation of fish [19–21]. Since otolith morphology has multiple purposes in fisheries, it is urgent to determine the factors that affect otolith variability.

Otolith morphology is generally affected by biotic (e.g., genetics, fish size, sex, and ontogeny) [22–24] and abiotic (e.g., temperature, pH, habitat preference, and food quality or quantity) [25–29] factors. Cardinale et al. [22] reported that even under the same growth conditions, the otolith morphologies of different populations of salmonids were different, which was the result of genetic effects [30]. Fey and Greszkiewicz [29] verified that the otolith size of *Esox lucius* was positively correlated with fish size, and Strelcheck et al. [31] demonstrated that slower-growing *Mycteroperca microlepis* had larger, heavier otoliths than equal-sized faster-growing *M. microlepis*. These studies indicate that the factors affecting otolith morphology are quite complex. However, previous studies on otolith morphology and its determinants mostly focused on marine fish [32,33], estuarine fish [17,34], and riverine fish [35]. There has been little research on lacustrine fish, especially in high-altitude lakes.

The Qinghai-Tibet Plateau (QTP), known as the roof of the world, has the highest altitude worldwide. It is the cradleland of major rivers in China; it also possesses the highest-altitude lake group in the world. Despite a few decades of investigation, a large number of lakes on the QTP still lack detailed fishing resource information. Lake Amdo Tsonak, located at the headwaters of the Salween River, is a typical high-altitude lake on the QTP. However, fishing resource information for this important and representative lake has rarely been reported. The species *Schizopygopsis thermalis* Herzenstein 1891 (Cyprinidae: Schizotritinae) is endemic to the QTP of China. It has the common characteristics of schizothoracine species: restricted distributions, a slow growth rate, and late sexual maturity as a result of adapting to harsh environments [36,37]. Two discrete intraspecific morphs of *S. thermalis*, planktivorous and benthivorous, were confirmed by Qiao et al. [38] in Lake Amdo Tsonak Co. They show differences in external morphological characteristics, feeding habits, and habitat preferences. However, the otolith-fish size relationship of *S. thermalis* is unclear; whether otolith morphology differs significantly between the two morphs is also unclear. Solving these problems could help avoid inconveniences in important fishery resource assessment and management. Thus, the specific aims of this study were to (1) quantify the relationships between otolith morphological measurements and fish length, and (2) measure otolith morphological differences between the two morphs of *S. thermalis* in Lake Amdo Tsonak Co.

## 2. Materials and Methods

### 2.1. Study Area and Field Sampling

Lake Amdo Tsonak Co (31.55–32.08° N, 91.25–91.33° E) is an oligotrophic, high-altitude and low-temperature freshwater lake of the headwaters of the Salween River (Nujiang) on the QTP. Since this research was conducted based on our previous field investigation on Lake Amdo Tsonak Co, this paper does not provide a specific description of the study area. Please see our study published in 2020 for details [38]. A total of 435 catch samples were captured in Lake Amdo Tsonak Co and its tributary (Nagchu River) (Figure 1) in May and September 2017, and April and July 2018 with gill nets and cast nets. After we

measured the standard length (SL, 0.1 mm), total length (TL, 0.1 mm), and total weight (*W*), and recorded the sex of each specimen in the field (https://www.koaw.org/, accessed on 10 April 2022), we extracted otoliths (left and right) from the utricle of all specimens with the help of fine forceps, cleaned them using distilled water to remove any additional membranes or surface residues, air-dried them, and stored them in labeled plastic tubes, and then weighed them using an electronic balance (AR1140, Ohaus Corporation) in the laboratory. For basic information about the studied samples, please see Table 1. All state and institutional guidelines for the care and use of animals were followed in this research.

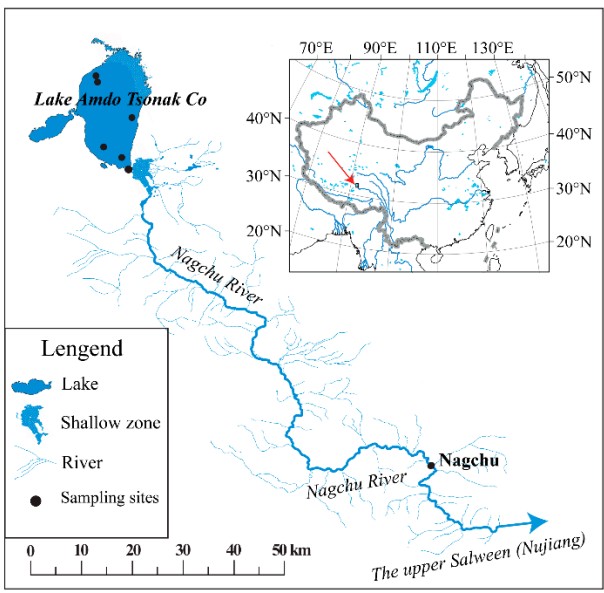

**Figure 1.** *Schizopygopsis thermalis* sampling locations (black dots) in Lake Amdo Tsonak Co. Blue arrows represent the direction of the stream.

**Table 1.** Summary of basic information of two morphs of *Schizopygopsis thermalis* collected in 2017 and 2018, including total length (TL) and weight with standard deviation (SD), TL groups and sex ratio.

| | Morphs | N | Total Length (mm) | | Weight (g) | | Sex Ratio |
|---|---|---|---|---|---|---|---|
| | | | Range | Mean ± SD | Range | Mean ± SD | Male:Female |
| All | Benthivorous morph | 206 | 268–517 | 372.35 ± 52.67 | 109.2–1036.2 | 440.88 ± 181.49 | 2.07:1 |
| | Planktivorous morph | 226 | 220–515 | 383.52 ± 56.70 | 80.1–974.6 | 450.44 ± 177.87 | 1.86:1 |
| Group 1 | Benthivorous morph | 34 | 268–323 | 293.97 ± 14.83 | 109.2–280.8 | 204.38 ± 42.07 | 3.25:1 |
| | Planktivorous morph | 37 | 220–321 | 292.41 ± 18.98 | 80.1–280.0 | 201.47 ± 36.57 | 3.6:1 |
| Group 2 | Benthivorous morph | 123 | 324–410 | 366.04 ± 25.48 | 238.9–706.4 | 413.42 ± 99.86 | 1.37:1 |
| | Planktivorous morph | 111 | 324–410 | 372.33 ± 24.91 | 140.4–649.4 | 404.71 ± 90.82 | 0.85:1 |
| Group 3 | Benthivorous morph | 49 | 412–517 | 442.59 ± 23.57 | 412.4–1036.2 | 673.94 ± 130.85 | 6:1 |
| | Planktivorous morph | 78 | 411–515 | 442.64 ± 25.09 | 409.0–974.6 | 633.63 ± 115.07 | 6.09:1 |

For marine fish, the sagittal otoliths are the largest of the three pairs of otoliths, and they are more commonly used in otolith morphology studies than the lapillus otoliths and asteriscus otoliths [39]. For freshwater Cyprinidae fishes, especially schizothoracine fishes, the sagittal otoliths are difficult to extract and measure due to their special aciculiform shape, so they are usually not suitable for morphological analysis. Lapillus otoliths of schizothoracine fishes have moderate sizes and stable shapes, and are thus mainly used for studying life history [40,41]. Thus, lapillus otoliths (Figure 2) were the first choice for measuring otolith morphology in this research [42,43].

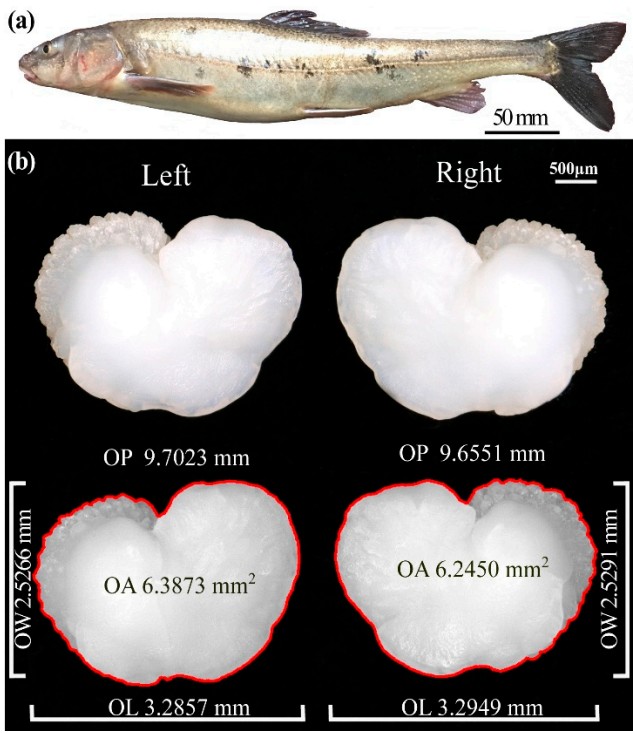

**Figure 2.** (**a**) Representative *Schizopygopsis thermalis* specimen; (**b**) three-dimensional structure of the left and right otoliths; red outlines indicate the two-dimensional contour of otoliths. OL: otolith length; OW: otolith width; OA: otolith area; OP: otolith perimeter.

### 2.2. Total Length-Based Group Divisions

To compare the otolith morphology of the two morphs of *S. thermalis*, all samples of planktivorous (N = 206) and benthivorous (N = 226) morphs were divided into three TL classes (Table 1). The demarcation criterion was that the TL frequency distributions of the two morphs' samples in each group were very close to each other to reduce the error caused by individual size differences in otolith morphological analysis. To explore how otolith measurements varied with TL, the relationship between each morph otolith measurement and TL was fitted for each length group. This research presents a comparative analysis of the otolith morphology of the two morphs in each length group to determine whether otolith morphology differed significantly between the morphs.

### 2.3. Otolith Morphometry

Before photographing the otoliths, they were placed consistently on a microscope stage covered with a black light-absorbing cloth to reduce light reflection and ensure true measurements of otolith morphology, where the sulcus side was perpendicular to the microscope stage and the anterior (rostral) region was facing up. Orthogonal two-dimensional digital images of each otolith were captured using a new type of three-dimensional (3D) color microscope (VHX 5000, Keyence, Osaka, Japan). The otolith outline was quantified by wavelet coefficients, which are preferred over Fourier coefficients for detecting differences in shape in specific regions of the otolith, using ShapeR [44] in R [45]. For each otolith, we measured otolith area (OA, in mm$^2$), otolith perimeter (OP, in mm), otolith length (OL, in mm), and otolith width (OW, in mm) (Figure 2). Otolith shape indices, including form-factor ($4\pi OA/OP^2$), circularity ($OP^2/OA$), rectangularity ($OA/[OL \times OW]$), ellipticity ($[OL-OW]/[OL+OW]$), roundness ($4OA/\pi OL^2$), aspect ratio ($OL/OW$) and surface density (OWE/OA), were then calculated based on the equations proposed by Tuset et al. [44].

## 2.4. Statistical Analysis

Fish size usually affects the morphological characteristics of otoliths [29,46]. To eliminate the allometric effect on otolith size as a consequence of variation in fish size, we calculated the allometric relationship between each otolith size characteristic (area, perimeter, length, and width) and fish TL to standardize our measurements to standard body size. We used the standard equation $Y = aX^{b_i}$ (where $b_i$ is the allometry parameter for each morph) and used logarithmic transformation to homogenize the residuals for fitting [2,35]. Therefore, each otolith size value $Y_{ij}$ was transformed into $Z_{ij}$ using the following formula [35,47]:

$$Z_{ij} = Y_{ij} \left[ X_0 / X_j \right]^{b_i}$$

where i is the untransformed otolith measurement value for the jth fish, $X_j$ is the TL of the jth individual, $X_0$ is the mean TL value of all individuals, and $b_i$ is the allometry parameter relating the dependent variable $Y_i$ to the independent variable $X_i$. $Z_{ij}$ is the theoretical value of $Y_{ij}$.

The shape indices were tested for normality and homogeneity using the Shapiro–Wilk normality test ($n < 50$ samples), Kolmogorov–Smirnov normality test ($n > 50$ samples), and Levene's test, respectively. The shape indices were log(x+1) transformed if they did not adhere to a normal distribution; data that could not be normalized or homogenized via transformation were excluded from further analyses. A correlation matrix was used to verify pairwise correlations between shape indices to examine possible multicollinearity. The shape indices that showed multicollinearity ($r > 0.7$) were eliminated [48]. Canonical analysis of principal coordinates (CAP) and pairwise t-tests revealed no significant differences in outlines and shape indices between the left and right otoliths, respectively (CAP, $F = 0.0806$; all $p > 0.05$). Thus, we measured the left otolith when available, and the right otolith if the left was not utilizable or absent. The chi-square test was used to assess whether the sex ratio differed significantly between morphs within each TL group. Since we were most interested in exploring whether otolith morphology differed significantly between the two morphs of *S. thermalis*, if the sex ratio was not significantly different between the two morphs in each TL group, the individuals of both sexes were combined for subsequent analysis. Note that this did not prevent us from testing whether there is sexual dimorphism in the otolith morphology of each morph. Student's *t*-test was implemented to evaluate whether the otolith shape indices differed significantly between sexes in each morph within each TL group. The Student's *t*-test was also performed to detect significant differences in otolith shape indices between morphs. Finally, the relationships between TL and the otolith measurements (e.g., OL, OW, OA, OP, and OWE) were determined by linear regression, in which all variables were log-transformed [24].

## 3. Results

### 3.1. Length Frequency Distributions of Fish Samples

A total of 432 specimens of *S. thermalis* were used in this research. The specimens were divided into three TL groups (Table 1). Basic information including the number of individuals of each morph, TL, weight, and sex ratio of specimens is shown in Table 1. There was no significant difference (all $p > 0.05$) in total fish length between the benthivorous and planktivorous morphs in each group, which meant that the TL in each group was approximately normally distributed (Figure 3). The median-based homogeneity test indicated that the TL data of each TL group satisfied the assumption of homogeneity of variances (all $p > 0.05$).

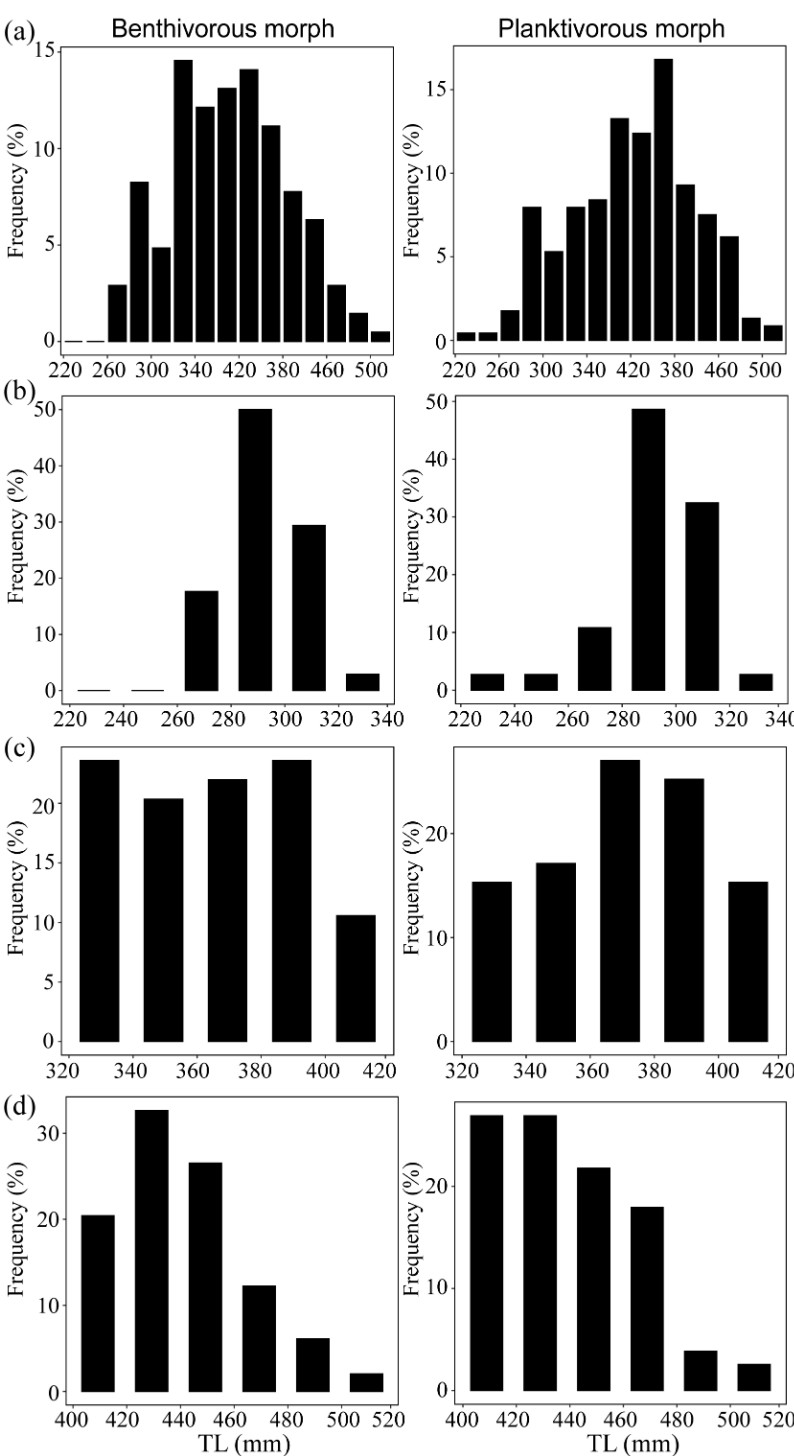

**Figure 3.** Length–frequency distribution (%) for benthivorous and planktivorous morphs of *Schizopy-gopsis thermalis* used in the otolith shape analysis: (**a–d**) represent the combination of all TL groups, first TL group, second TL group, and third TL group, respectively.

*3.2. Relationships between Shape Indices and TL*

For both morphs in all TL groups, OL, OW, OA, OP, and OWE were linearly positively correlated with TL. In other words, with an increase in TL, these otolith measurements also increased. (Figure 4). The slope and coefficient of determination ($R^2$) of the linear relationships between TL and otolith measurements ranged from 0.516–2.53 and 0.09–0.73, respectively (Table 2). All relationships between TL and otolith measurements were sta-tistically significant (all $p < 0.05$) (Table 2; Figure 4). The relationships between TL and

otolith measurements of all TL groups combined, TL group 1, TL group 2, and TL group 3 in each morph displayed high, moderate, slightly low, and low coefficients of determination, respectively. In addition, the planktivorous morph showed a much higher coefficient of determination and regression value (slope) for the relationships between TL and otolith measurements (except between TL and OW) than the benthivorous morph (Table 2; Figure 4).

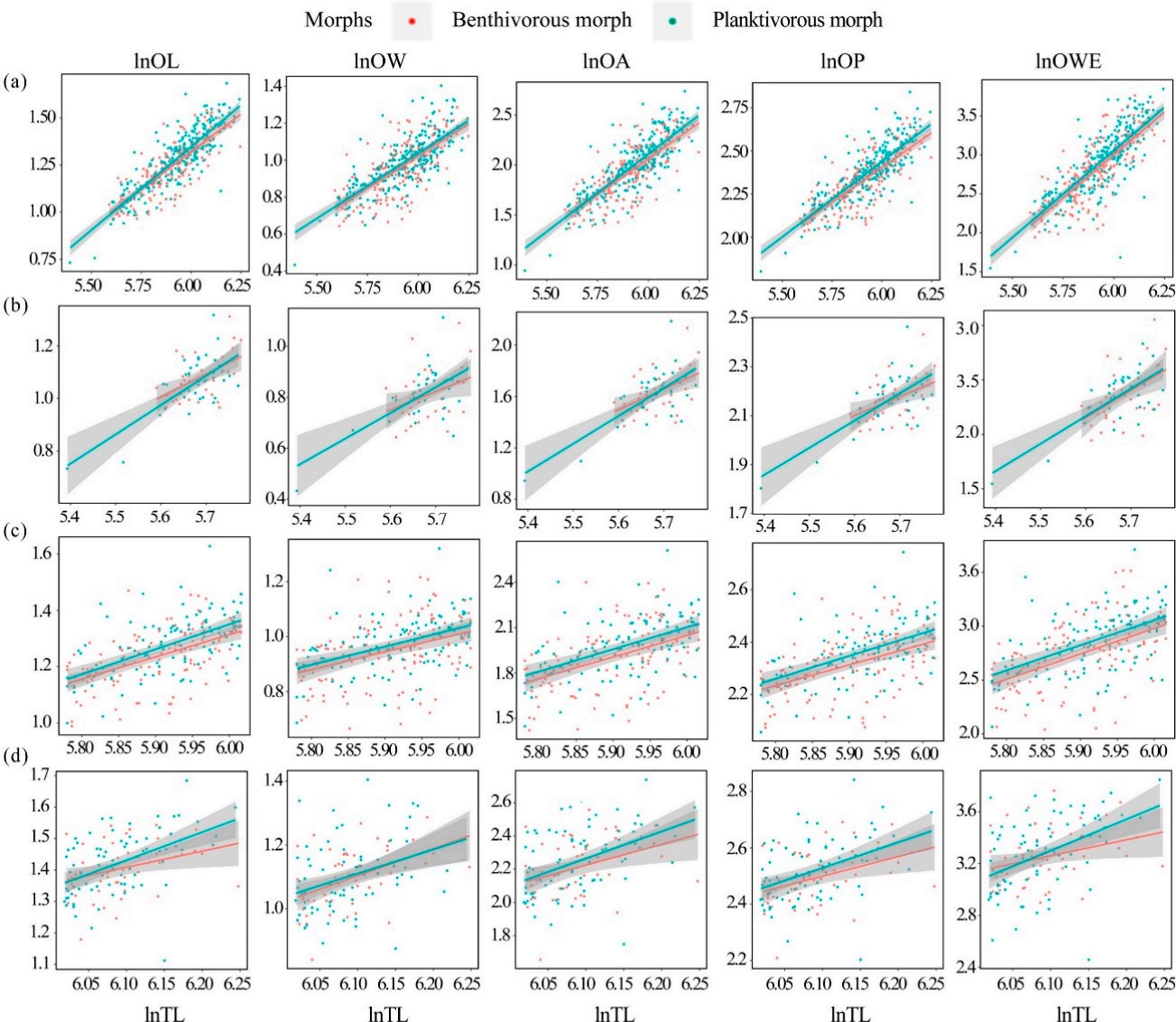

**Figure 4.** Relationships of total fish length with the otolith shape indices according to *Schizopygopsis thermalis* captured from Lake Amdo Tsonak Co. Redpoint and linear equation belong to benthivorous morph while green point and linear equation pertain to planktivorous morph. (**a–d**) represent the combination of all TL groups, first TL group, second TL group, and third TL group, correspondingly.

The average otolith shapes of the two morphs in all TL groups were reconstructed based on wavelet transformation (Figure 5). For all TL groups combined, morphological differences in otoliths of the two morphs were mainly observed in specific regions, especially in the rostrum, excisura ostii, posterior side, and region between the posterior and ventral otolith (CAP, F = 3.4187, $p < 0.05$; Figure 5a). This was consistent with the variability in the means and standard deviations of wavelet coefficients. The intraclass correlation coefficient explained a large proportion of the intraspecific variation (Figure 5a) and revealed a high level of intermorph variation in the wavelet coefficients for the mean outline at 30–90°,

270–300°, and 320–360° (Figure 5a). Similarly, concerning TL group 2, the otolith outlines of the two morphs mainly differed in specific regions, i.e., the posterior side and the region between the posterior and ventral otolith (CAP, F = 2.2299, $p < 0.05$; Figure 5c). The intraclass correlation coefficient revealed a moderate level of intermorph variation in the wavelet coefficients for the mean outline at 240–280° and 320–340° (Figure 5c). However, the otolith outlines of the two morphs of both TL group 1 and TL group 3 were not significantly different (CAP, all $p > 0.05$), even though the mean otolith shape appeared to overlap less between the two morphs (Figure 5b,d).

**Table 2.** The relationship between total fish length (TL) and shape indices (OL, OW, OA, OP, and OWE) (after log transformed) of otolith in the combination of all TL groups, first TL group, second TL group, and third TL group, respectively. Note: a: intercept; b: slope; $R^2$: coefficient of determination, *: $p < 0.05$, **: $p < 0.01$, and ***: $p < 0.001$.

| | Independent Variables | Dependent Variables | Morphs | Equation | a | b | $R^2$ | p |
|---|---|---|---|---|---|---|---|---|
| **All** | TL | OL | Benthivorous morph | y = −3.45 + 0.795x | −3.45 | 0.8 | 0.67 | <0.001 *** |
| | | | Planktivorous morph | y = −3.95 + 0.883x | −3.95 | 0.88 | 0.73 | <0.001 *** |
| | TL | OW | Benthivorous morph | y = −3.19 + 0.702x | −3.19 | 0.7 | 0.49 | <0.001 *** |
| | | | Planktivorous morph | y = −3.21 + 0.708x | −3.21 | 0.71 | 0.58 | <0.001 *** |
| | TL | OA | Benthivorous morph | y = −6.55 + 1.44x | −6.55 | 1.44 | 0.59 | <0.001 *** |
| | | | Planktivorous morph | y = −7.22 + 1.55x | −7.22 | 1.55 | 0.68 | <0.001 *** |
| | TL | OP | Benthivorous morph | y = −2.42 + 0.804x | −2.42 | 0.8 | 0.61 | <0.001 *** |
| | | | Planktivorous morph | y = −2.80 + 0.873x | −2.80 | 0.87 | 0.68 | <0.001 *** |
| | TL | OWE | Benthivorous morph | y = −10.30 + 2.21x | −10.30 | 2.21 | 0.61 | <0.001 *** |
| | | | Planktivorous morph | y = −10.40+ 2.24x | −10.40 | 2.24 | 0.64 | <0.001 *** |
| **Group 1** | TL | OL | Benthivorous morph | y = −3.87 + 0.87x | −3.87 | 0.87 | 0.26 | =0.002 ** |
| | | | Planktivorous morph | y = −5.72 + 1.11x | −5.72 | 1.11 | 0.51 | <0.001 *** |
| | TL | OW | Benthivorous morph | y = −3.26 + 0.716x | −3.26 | 0.72 | 0.13 | =0.04 * |
| | | | Planktivorous morph | y = −4.91 + 1.01x | −4.91 | 1.01 | 0.41 | <0.001 *** |
| | TL | OA | Benthivorous morph | y = −7.48 + 1.60x | −7.48 | 1.6 | 0.23 | =0.004 ** |
| | | | Planktivorous morph | y = −10.60 + 2.16x | −10.60 | 2.16 | 0.51 | <0.001 *** |
| | TL | OP | Benthivorous morph | y = −2.24 + 0.775x | −2.24 | 0.78 | 0.2 | =0.009 ** |
| | | | Planktivorous morph | y = −4.15 + 1.11x | −4.15 | 1.11 | 0.46 | <0.001 *** |
| | TL | OWE | Benthivorous morph | y = −11.70 + 2.48x | −11.70 | 2.48 | 0.23 | =0.004 ** |
| | | | Planktivorous morph | y = −12.00 + 2.53x | −12.00 | 2.53 | 0.53 | <0.001 *** |
| **Group 2** | TL | OL | Benthivorous morph | y = −3.41 + 0.788x | −3.41 | 0.79 | 0.31 | <0.001 *** |
| | | | Planktivorous morph | y = −3.96 + 0.884x | −3.96 | 0.88 | 0.34 | <0.001 *** |
| | TL | OW | Benthivorous morph | y = −2.88 + 0.649x | −2.88 | 0.65 | 0.15 | <0.001 *** |
| | | | Planktivorous morph | y = −2.99 + 0.67x | −2.99 | 0.67 | 0.21 | <0.001 *** |
| | TL | OA | Benthivorous morph | y = −6.49+ 1.42x | −6.49 | 1.42 | 0.24 | <0.001 *** |
| | | | Planktivorous morph | y = −6.67 + 1.46x | −6.67 | 1.46 | 0.27 | <0.001 *** |
| | TL | OP | Benthivorous morph | y = −2.36 + 0.792x | −2.36 | 0.79 | 0.25 | <0.001 *** |
| | | | Planktivorous morph | y = −2.81 + 0.875x | −2.81 | 0.88 | 0.3 | <0.001 *** |
| | TL | OWE | Benthivorous morph | y = −11.20 + 2.36x | −11.20 | 2.36 | 0.28 | <0.001 *** |
| | | | Planktivorous morph | y = −10.90 + 2.32x | −10.90 | 2.32 | 0.29 | <0.001 *** |
| **Group 3** | TL | OL | Benthivorous morph | y = −1.74 + 0.516x | −1.74 | 0.52 | 0.11 | =0.021 * |
| | | | Planktivorous morph | y = −4.01 + 0.891x | −4.01 | 0.89 | 0.25 | <0.001 *** |
| | TL | OW | Benthivorous morph | y = −4.06 + 0.846x | −4.06 | 0.85 | 0.22 | <0.001 *** |
| | | | Planktivorous morph | y = −3.50 + 0.756x | −3.50 | 0.76 | 0.14 | <0.001 *** |
| | TL | OA | Benthivorous morph | y = −5.41 + 1.25x | −5.41 | 1.25 | 0.14 | =0.009 ** |
| | | | Planktivorous morph | y = −7.65 + 1.63x | −7.65 | 1.63 | 0.22 | <0.001 *** |
| | TL | OP | Benthivorous morph | y = −1.75 + 0.697x | −1.75 | 0.7 | 0.14 | =0.007 ** |
| | | | Planktivorous morph | y = −3.00 + 0.907x | −3.00 | 0.91 | 0.2 | <0.001 *** |
| | TL | OWE | Benthivorous morph | y = −4.26 + 1.23x | −4.26 | 1.23 | 0.09 | =0.038 * |
| | | | Planktivorous morph | y = −11.40 + 2.41x | −11.40 | 2.41 | 0.21 | <0.001 *** |

After multicollinearity diagnosis, the shape indices OL, aspect ratio, circularity, and surface density were retained in the combination of all TL groups, TL group 1, and TL group 2 for subsequent analyses because they conveyed more otolith morphology information than the other shape indices. For TL group 3, OL, OW, aspect ratio, circularity, and surface density were reserved. Student's $t$-test revealed that otolith morphology exhibited sexual dimorphism in each morph of *S. thermalis* (Table 3). Circularity and surface density displayed sexual dimorphism in both morphs in group 1 and all TL groups combined, respectively; circularity and aspect ratio exhibited sexual dimorphism only in the planktivorous morph in TL group 2 and all TL groups combined, and surface density displayed sexual dimorphism only in the benthivorous morph in TL group 2 and TL group 3 (Table 3). As the chi-square test showed that the sex ratio was not significantly different between the two morphs in each TL group (all TL groups: $\chi^2_{(1, n = 432)} = 0.285$; first TL group: $\chi^2_{(1, n = 71)} = 0.037$;

second TL group: $\chi^2_{(1, n = 234)} = 3.243$, and third TL group: $\chi^2_{(1, n = 127)} = 0.001$; all $p > 0.05$), and few shape indices exhibited sexual dimorphism within morphs in our research, the individuals of both sexes within morphs were merged to explore differences in otolith morphology between the two morphs. In both TL group 2 and the combination of all TL groups, Student's *t*-test revealed that OL, circularity, and surface density were significantly different between the two morphs (all $p < 0.05$, Table 4). OL, circularity, and surface density in the planktivorous morph were significantly higher than those in the benthivorous morph (Table 4). However, no significant difference was observed in any otolith shape index between the two morphs in TL group 1 and TL group 2 (all $p > 0.05$, Table 4).

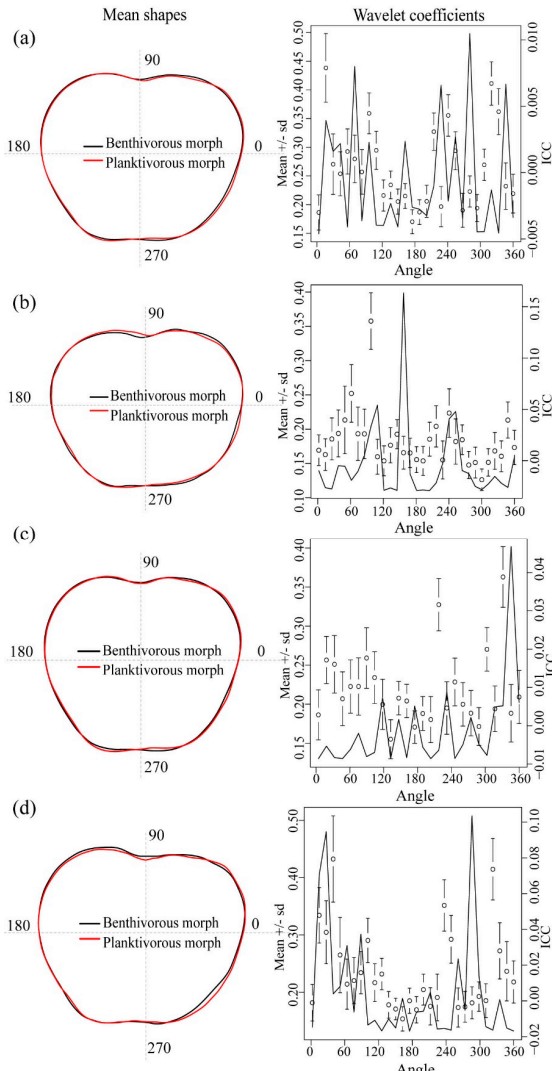

**Figure 5.** Red and black contours indicate mean otolith shapes, based on wavelet reconstruction for planktivorous and benthivorous morphs of *Schizopygopsis thermalis*. Line chart signifies means and standard deviations of the wavelet coefficients (black circles) which represent the mean shape of all otoliths, and the magnitude of variation in *S. thermalis* (the intraclass correlation coefficient, black line). Both x-axes show angles in degrees (°) based upon polar coordinates, where the centroid of the otolith is the center point of the polar coordinates. (**a**–**d**) denote the combination of all TL groups, first TL group, second TL group, and third TL group, correspondingly.

**Table 3.** Results of Student's *t*-test of sexes of each morph based on otolith shape indices which contained the combination of all TL groups, first TL group, second TL group, and third TL group, respectively. *: $p < 0.05$, **: $p < 0.01$ and ***: $p < 0.001$.

| | Shape Indices | Benthivorous Morph | | | Planktivorous Morph | | |
|---|---|---|---|---|---|---|---|
| | | *t* | df | *p* | *t* | df | *p* |
| All | Otolith length | −1.923 | 204 | 0.056 | 1.422 | 224 | 0.156 |
| | Aspect ratio | 0.872 | 204 | 0.384 | 2.036 | 224 | 0.043 * |
| | Circularity | −1.512 | 204 | 0.132 | −0.439 | 224 | 0.661 |
| | Surface density | −4.15 | 204 | 0.000 *** | −2.815 | 224 | 0.005 ** |
| Group 1 | Otolith length | −1.622 | 32 | 0.115 | −0.937 | 35 | 0.355 |
| | Aspect ratio | 1.885 | 32 | 0.069 | 1.152 | 35 | 0.257 |
| | Circularity | −2.168 | 32 | 0.038 * | −2.316 | 35 | 0.027 * |
| | Surface density | −0.17 | 32 | 0.866 | 0.51 | 35 | 0.614 |
| Group 2 | Otolith length | −1.434 | 121 | 0.154 | 1.249 | 109 | 0.214 |
| | Aspect ratio | 0.376 | 121 | 0.707 | 0.983 | 109 | 0.328 |
| | Circularity | −0.17 | 121 | 0.865 | 2.71 | 109 | 0.008 ** |
| | Surface density | −4.195 | 121 | 0.000 *** | −0.914 | 109 | 0.363 |
| Group 3 | Otolith length | −0.02 | 47 | 0.984 | 1.335 | 76 | 0.186 |
| | Otolith width | 0.369 | 47 | 0.714 | 0.345 | 76 | 0.731 |
| | Aspect ratio | −0.504 | 47 | 0.617 | 0.876 | 76 | 0.384 |
| | Circularity | −0.437 | 47 | 0.664 | −0.493 | 76 | 0.623 |
| | Surface density | −2.034 | 47 | 0.048 * | −0.951 | 76 | 0.345 |

**Table 4.** Results of Student's *t*-test of two morphs of *Schizopygopsis thermalis* based on otolith shape indices which contained the combination of all TL groups, first TL group, second TL group, and third TL group, respectively. *: $p < 0.05$, **: $p < 0.01$ and ***: $p < 0.001$.

| | Shape Indices | Benthivorous Morph | Planktivorous Morph | *t* | df | *p* |
|---|---|---|---|---|---|---|
| All | Otolith length | 3.57 ± 0.29 | 3.64 ± 0.30 | −2.613 | 430 | 0.009 ** |
| | Aspect ratio | 1.34 ± 0.09 | 1.35 ± 0.09 | −0.798 | 430 | 0.425 |
| | Circularity | 15.42 ± 0.84 | 15.75 ± 0.99 | −3.672 | 430 | 0.000 *** |
| | Surface density | 2.43 ± 0.83 | 2.65 ± 0.95 | −2.545 | 430 | 0.011 * |
| Group 1 | Otolith length | 2.94 ± 0.22 | 2.91 ± 0.22 | 0.578 | 69 | 0.565 |
| | Aspect ratio | 1.31 ± 0.07 | 1.28 ± 0.09 | 1.327 | 69 | 0.189 |
| | Circularity | 14.91 ± 0.52 | 15.17 ± 0.75 | −1.713 | 69 | 0.091 |
| | Surface density | 2.11 ± 0.36 | 2.13 ± 0.40 | −0.225 | 69 | 0.823 |
| Group 2 | Otolith length | 3.48 ± 0.29 | 3.58 ± 0.31 | −2.57 | 232 | 0.011 * |
| | Aspect ratio | 1.34 ± 0.09 | 1.35 ± 0.07 | −0.928 | 232 | 0.354 |
| | Circularity | 15.24 ± 0.83 | 15.51 ± 0.72 | −2.681 | 232 | 0.008 ** |
| | Surface density | 2.33 ± 0.48 | 2.48 ± 0.49 | −2.489 | 232 | 0.014 * |
| Group 3 | Otolith length | 4.09 ± 0.31 | 4.17 ± 0.35 | −1.395 | 125 | 0.165 |
| | Otolith width | 3.01 ± 0.25 | 3.03 ± 0.32 | −0.518 | 125 | 0.606 |
| | Aspect ratio | 1.36 ± 0.09 | 1.38 ± 0.10 | −1.066 | 125 | 0.289 |
| | Circularity | 16.04 ± 0.99 | 16.36 ± 1.35 | −1.417 | 125 | 0.159 |
| | Surface density | 2.85 ± 0.43 | 2.81 ± 0.59 | 0.332 | 125 | 0.740 |

## 4. Discussion

### 4.1. Relationships between Shape Indices and TL

The strong correlation between otolith morphology and fish size is common in marine fishes, such as *Merluccius capensis* [47], *Terapon jarbua* [24], and *Neogobius melanostomus* [49,50]. However, this relationship has been very poorly studied in freshwater fishes. Hence, basic data on the biology and dynamics of *S. thermalis* are essential for successful stock assessment and consequently, for fishing management in Lake Amdo Tsonak Co. In this research, OL showed a linear relationship with TL displaying the largest coefficient of

determination ($R^2$) among the shape indices in each TL group, which indicated that for *S. thermalis*, a better prediction of fish size could be obtained when OL information was available. However, OP, OWE, OA, and OW are still valuable predictors of fish size. Such a phenomenon also exists in *Cynoscion guatucupa* [51] and *T. jarbua* [24]. From TL group 1 to TL group 3, the slopes and coefficients of determination of each shape index exhibited high variability (Table 2; Figure 4), which could be explained by ontogenetic factors [52,53]. Furthermore, the planktivorous morph displayed stronger positive linear relationships between the shape indices and TL (except between OW and TL) in all TL groups than the benthivorous morph (Table 2; Figure 4). These results can be attributed to differences in habitat preferences, food quality, and growth features between the two morphs. In previous research, we demonstrated that the planktivorous morph and benthivorous morph of *S. thermalis* showed low dietary overlap, different growth rates, and habitat preferences [38]. The former morph predominantly fed on plankton and inhabited pelagic lake habitats, while the latter mainly fed on periphytic algae, and dwelled in the benthic zone and the tributaries of the lake; the former also exhibited a slower growth rate than the latter [38]. Research on the effects of diet, growth, and habitat preferences on the otolith-fish size relationship has also been performed in *Mullus microlepis* [31,54].

### 4.2. Otolith Morphometry

Sexual dimorphism in otolith morphology has been reported in various fish [23,55,56]. Sexual dimorphism can result from phenotypic plasticity induced by unequal growth rates, sex-specific hormone levels, and courtship behavior [23,39,55]. Qiao et al. [38] reported that growth characteristics, including asymptotic SL and growth rate, differed significantly between females and males within morphs, which could account for the sexual dimorphism of otolith morphology within morphs.

Otolith morphology (otolith shape and size) was significantly different between the benthivorous and planktivorous morphs in TL group 2 but not in TL group 1 or TL group 3. These results can be explained by the effects of ontogeny and environmental conditions on otolith morphology. In previous research, the polymorphic forms of *S. thermalis* were inferred to be an adaptation to differentiation in feeding biology and habitat utilization, as the two morphs of *S. thermalis* were confirmed to share a recent ancestor and a common pool of genetic variation [38]. Little obvious morphological variation in the juveniles was found in *S. thermalis* in a field investigation. Hence, no significant difference in otolith morphology was found between the two morphs in TL group 1, which could be interpreted as a response-hysteresis effect of otoliths, although the morphs had experienced habitat utilization differentiation [57,58]. The growth rates of fish and otoliths depend not only on the recent growth rate but also on growth inertia due to past growth [59]. The strongly autoregressive nature of otolith growth leads to inertia in growth processes that buffers the otolith from exogenous influences, and induces a lag between otolith response and environmental perturbations [57,60]. This makes it unlikely that shifts in habitat use will rapidly affect the ontogenetic direction of otolith development. The most direct evidence is that when the SL of *S. thermalis* is less than approximately 280 mm (approximately 320 mm TL), the SL-age relationships of the two morphs are consistent [38].

However, otolith morphology was significantly different between the two morphs in TL group 2, after the period with an otolith response-hysteresis effect of the environment had passed. Multiple factors can explain this result. Many studies have shown that habitat use and feeding behavior can significantly affect otolith shape [22,52,58]. The planktivorous morph inhabits the pelagic area and prefers feeding on animals, while the benthivorous morph dwells in the benthic zone on the shore of the lake and its tributaries, and prefers a plant-based diet [38]. In addition, various environmental factors, such as temperature [29], depth [26], and feeding conditions [25,31], are thought to affect fish growth, which in turn can influence otolith growth, thus producing variations in otolith shape in the absence of genetic differences [27,61]. Our results are similar to these previously reported results. In this paper, the otolith measurements showed a higher regression value

(slope) with TL in the planktivorous morph than in the benthivorous morph (Table 2; Figure 4), and a previous study demonstrated that the individuals of the planktivorous morph exhibited a larger asymptotic SL ($L_\infty$ = 405.14) than individuals of the benthivorous morph ($L_\infty$ = 374.22) [38]. Thus, otolith morphology differences between two morphs are related to growth features' differences between the two morphs [31,61]. Otoliths are calcareous structures in the inner ear and have movement, auditory, and other sensory functions in fish [3,10]. Schulz-Mirbach et al. [3] reported that otolith shape and mass were associated with various locomotion behaviors in fish, such as body tilts, swimming (acceleration) along the rostrocaudal or dorso-ventral axes, and movements from left to right [62,63]. The planktivorous morph inhabiting the pelagic area possessed a more complex (e.g., circularity), larger (OL), and denser (surface density) otolith (Table 4) than the benthivorous morph that dwells in the benthic zone and tributaries of the lake. The planktivorous morph with a more complex otolith shape may have higher mobility [13], such as swimming and orientating rapidly for predation (e.g., zooplankton and small fishes), and avoiding predators (e.g., bird peck wounds were observed on the body surface of some planktivorous morph individuals). Regarding microstructure, otoliths are composed of $CaCO_3$ that normally precipitates as aragonite. The difference in the deposition of carbonate (surface density) between the two morphs may be related to diet differentiation [64].

Many studies have shown that the adaptation of otolith growth to environmental changes has a lag effect and inertia [57,59,60], but once adapted, shape divergence increases with age/size [58]. However, the otolith morphology differentiation of *S. thermalis* observed in this study may be a special case among fish. No significant difference in otolith shape or size was detected in TL group 3. This counterintuitive result may be related to the biological attributes of *S. thermalis*. Previous studies have shown that otolith shape changes in different regions are not obvious due to the slow growth of deep-water fish [65,66]. *S. thermalis* lives in an oligotrophic, high-altitude, and low-temperature lake, Lake Amdo Tsonak Co, on the QTP; its slow growth rate (>26-year-old individuals can reach an SL of 380 mm) is an adaptation to this harsh environment [38]. Hence, a slow growth rate may impede the divergence of otolith growth trajectories and reduce intraspecific variation in otolith shape. This phenomenon has also been reported in the naked carp, *Gymnocypris selincuoensis*, on the QTP [43].

## 5. Conclusions

In conclusion, *S. thermalis* exhibited a strong linear correlation between otolith morphology and fish size, and displayed intraspecific variation in otolith morphology (e.g., otolith shape and size). In conjunction with our previous study [38], this study also shows that the otolith morphology of *S. thermalis* is related to biotic (e.g., growth features) and abiotic (e.g., diet and habitat preference) factors. Basic data on the biology of *S. thermalis* are essential for poorly studied Lake Amdo Tsonak Co, and our study emphasizes that intraspecific variation in otolith morphology should be taken into consideration when differentiating stocks, populations, and age classes based on otolith morphology. However, due to extreme environmental conditions and the endemism of *S. thermalis*, we are unable to study *S. thermalis* in great detail by conducting long-term control experiments in high-altitude areas or by simulating high-altitude lake environmental conditions in laboratories located at lower altitudes; long-term fishery resource monitoring for *S. thermalis* in high-altitude lakes is also rather difficult. Otolith microchemistry should be used in future studies to explore the entire life history of *S. thermalis*.

**Author Contributions:** Conceptualization, J.Q., Y.Y., and D.H.; Data curation, R.Z. and K.C.; Formal analysis, J.Q.; Investigation, R.Z., K.C., and D.H.; Methodology, D.Z.; Resources, Y.Y. and D.H.; Software, D.Z.; Writing—original draft, J.Q.; Writing—review and editing, J.Q. and D.H. All authors have read and agreed to the published version of the manuscript.

**Funding:** This project was supported by the Second Tibetan Plateau Scientific Expedition Program (No. 2019QZKK05010102) and National Natural Science Foundation of China (No. 32070436).

**Institutional Review Board Statement:** All experimental protocols were approved by the Ethics Committee of the Institute of Hydrobiology, Chinese Academy of Sciences. (Approval code: IHB/LL/2017027, date: 1 April 2017).

**Data Availability Statement:** The data of the present study are available from the authors upon reasonable request.

**Acknowledgments:** We are thankful to Yintao Jia and Xiu Feng for field sampling.

**Conflicts of Interest:** The authors declare no conflict of interest.

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
