# Peer review of "Comparative Otolith Morphology of Two Morphs of Schizopygopsis thermalis Herzenstein 1891 (Pisces, Cyprinidae) in a Headwater Lake on the Qinghai-Tibet Plateau"

_fishes, doi:10.3390/fishes7030099_

Round 1

Reviewer 1 Report

The manuscript id "fishes-1697823"is very interesting and create the basic knowledge regarding this fish species living a peculiar habitat. 

I have the following suggestions to improve the quality of the manuscript:

  • Please avoid in keywords the use of terms already used in the title
  • Please be sure to add species authorities each first time a species is mentioned
  • Line 66: reference [30] is about salmonids not about Gadus morhua 
  • Lines 70-71:  "However, previous studies on otolith morphology and its determinants mostly focused on marine fish [4]" other papers on marine fishes otolith morphology and how it is related to environmental parameters and other factors are quite common especially in the last years; see for example: "Otolith Analyses Highlight Morpho-Functional Differences of Three Species of Mullet (Mugilidae) from Transitional Water, D'Iglio et al., 2022, Sustainability" and "Intra- and interspecific variability among congeneric Pagellus otoliths, D'Iglio et al., 2021, Scientific Reports"
  • Figure 1 should be of a better quality to better locate the sampling points
  • When a sentence begins with a scientific name please avoid to punctate the name of the genus. 
  • As the authors choose lapillus for their study it would be better to use this term than otoliths that indicate all the three structures generically.
  • Why is the first part of results written in italics?

Author Response

Reviewer# 1:

Comments to the Author

The manuscript id "fishes-1697823"is very interesting and create the basic knowledge regarding this fish species living a peculiar habitat.

Authors’ Response

We are very grateful for the reviewer’s approval of our manuscript. We appreciate the suggestions made by the reviewer, and we have carefully addressed all the comments to improve our manuscript.

Comment 1
Please avoid in keywords the use of terms already used in the title

Authors’ Response

We have changed “otolith morphology” to “otolith shape”. (Revised MS Lines 27)

Comment 2
Please be sure to add species authorities each first time a species is mentioned

Authors’ Response

We have solved these types of problems in the whole text.

Comment 3:

Line 66: reference [30] is about salmonids not about Gadus morhua
Authors’ Response

We changed the “Gadus morhua” to “salmonids”. (Revised MS Lines 64)

Comment 4:
Lines 70-71:  "However, previous studies on otolith morphology and its determinants mostly focused on marine fish [4]" other papers on marine fishes otolith morphology and how it is related to environmental parameters and other factors are quite common especially in the last years; see for example: "Otolith Analyses Highlight Morpho-Functional Differences of Three Species of Mullet (Mugilidae) from Transitional Water, D'Iglio et al., 2022, Sustainability" and "Intra- and interspecific variability among congeneric Pagellus otoliths, D'Iglio et al., 2021, Scientific Reports"
Authors’ Response

We have added two new articles to enrich the research context here. (Revised MS Lines 70)

Comment 5:
Figure 1 should be of a better quality to better locate the sampling points
Authors’ Response

We have updated Figure 1. (Revised MS Lines 109)

Comment 6:
When a sentence begins with a scientific name please avoid to punctate the name of the genus. 
Authors’ Response

We have solved these problems in the whole text.

Comment 7:
As the authors choose lapillus for their study it would be better to use this term than otoliths that indicate all the three structures generically.
Authors’ Response

We appreciate and fully understand the suggestions made by the reviewer. However, in most research papers on otolith morphology, it is clearly stated in the method that the research object is one of the three types of otoliths (sagittal otoliths, lapillus otoliths, and asteriscus otoliths), and then otoliths are used as a general term for the research objects in the full text. In this manuscript, we explicitly use lapillus otoliths as the study object in the methods, so we want to keep the original term in the text.

Comment 8:
Why is the first part of results written in italics?

Authors’ Response

We realized that the sentences in the first part of the results were unbefitting; we have revised them. (Revised MS Lines 191-198)

Reviewer 2 Report

Review

Paper title: Comparative otolith morphology of two morphs of Schizopygopsis thermalis Herzenstein 1891 (Pisces, Cyprinidae) in a headwater lake on the Qinghai-Tibet Plateau

The authors conducted a comparative study to reveal the possible differences in the otolith morphology of two morphotypes of Schizopygopsis thermalis in Lake Amdo Tsonak Co. The authors measured and calculated otolith indices in 432 fish specimens and related the differences to planktivorous and benthivorous morphs as well as males and females. The authors found sex-specific variations and the role of lifestyle in determining the otolith morphology with the planktivorous morph having a more complex otolith shape due to higher activity. The authors concluded the presence of two distinct groups of otoliths corresponding to two different morphs. These results may have important implications for management and conservation purposes.

All these reasons explain the relevance of the paper by Jialing Qiao and co-authors submitted to "Fishes".

General scores.

The data presented by the authors are original and significant. The study is correctly designed and the authors used appropriate collecting methods. In general, the statistical analyses are performed with good technical standards. The authors conducted careful work that may attract the attention of a wide range of specialists focused on fish biology and fisheries management.

Recommendations.

Figures 1 and 5 are of low resolution. The authors should increase both the font size and resolution.  

L 102. The authors should specify the terms “standard length” and “total length” or provide a relevant citation for these terms.

Figure 3a, OX-axes. The authors should increase the font size

L 55-67. This section is redundant because it is a repetition of previously mentioned data and contains no discussion.

The authors declare that their data are important for the fisheries management of Schizopygopsis thermalis but they provide no information about the fishery potential of this species in Lake Amdo Tsonak Co. They should mention abundance, stock, and potential or real catch (if any) for this fish in the area.

References should be formatted according to Rules for Authors.

Specific comments.

P 1–7

L 14. Change “dynamics  in  fisheries” to “dynamics  of  fish”

L 74. Change “QTP” to “Qinghai-Tibet Plateau (QTP)”

L 80. Change “S. thermalis” to “Schizopygopsis thermalis

L 111. Change “S. thermalis” to “Schizopygopsis thermalis

L 124. Change “S. thermalis” to “Schizopygopsis thermalis

L 199. Change “morph” to “morphs”

L 199. Change “S. thermalis” to “Schizopygopsis thermalis

L 216. Change “according to the” to “according to”

L 216. Change “S. thermalis” to “Schizopygopsis thermalis

P 10-17

L 17. Change “they mean” to “the mean”

L 20. Change “morph” to “morphs”

L 20. Change “S. thermalis” to “Schizopygopsis thermalis

L 30. Change “The student’s t-test” to “Student’s t-test”

L 46. Change “student’s t test” to “Student’s t-test”

L 50. Change “student’s t test” to “Student’s t-test”

L 50. Change “S. thermalis” to “Schizopygopsis thermalis

L 72. Change “lakes” to “fishes”

L 73. Change “in fishing management” to “for fishing management”

L 134. Change “between  two  morphs” to “between the two  morphs”

Author Response

Reviewer(s)’ Comments to Author:

Reviewer# 2:

Comments to the Author
The authors conducted a comparative study to reveal the possible differences in the otolith morphology of two morphotypes of Schizopygopsis thermalis in Lake Amdo Tsonak Co. The authors measured and calculated otolith indices in 432 fish specimens and related the differences to planktivorous and benthivorous morphs as well as males and females. The authors found sex-specific variations and the role of lifestyle in determining the otolith morphology with the planktivorous morph having a more complex otolith shape due to higher activity. The authors concluded the presence of two distinct groups of otoliths corresponding to two different morphs. These results may have important implications for management and conservation purposes.

Authors’ Response

We are very grateful for the reviewer’s approval of our manuscript. We very much appreciate the suggestions made by the reviewer, and we have carefully addressed all the comments to improve our manuscript.

Comment 1:
Figures 1 and 5 are of low resolution. The authors should increase both the font size and resolution. 

Authors’ Response

We have updated Figures 1 and 5. (Revised MS Lines 109, 123, 199, 216, and 18)

Comment 2:
L 102. The authors should specify the terms “standard length” and “total length” or provide a relevant citation for these terms.

Authors’ Response

We appreciate the suggestion made by the reviewer and have added the relevant citation. (Revised MS Lines 102)

Comment 3:
Figure 3a, OX-axes. The authors should increase the font size

Authors’ Response

We have updated Figures 3. (Revised MS Lines 199)

Comment 4:
L 55-67. This section is redundant because it is a repetition of previously mentioned data and contains no discussion.

Authors’ Response

We agree with the reviewer’s comment and have deleted this part.

Comment 5:
The authors declare that their data are important for the fisheries management of Schizopygopsis thermalis but they provide no information about the fishery potential of this species in Lake Amdo Tsonak Co. They should mention abundance, stock, and potential or real catch (if any) for this fish in the area.

Authors’ Response

We appreciate and fully understand the suggestions made by the reviewer. We have added the abundance of Schizopygopsis thermalis in the “2.1. Study area and field sampling”. (Revised MS Lines 98-99)

Comment 6:
References should be formatted according to Rules for Authors.

Authors’ Response

According to Rules for Authors, we have revised some references that were not properly formatted.

Comment 7:
L 14. Change “dynamics in fisheries” to “dynamics of fish”

Authors’ Response

We have revised this problem. (Revised MS Lines 14)

Comment 8:
L 74. Change “QTP” to “Qinghai-Tibet Plateau (QTP)”

Authors’ Response

We have revised this problem. (Revised MS Lines 73)

Comment 9:

L 80. Change “S. thermalis” to “Schizopygopsis thermalis

Authors’ Response

We have revised this problem. (Revised MS Lines 79)

Comment 10:
L 111. Change “S. thermalis” to “Schizopygopsis thermalis

Authors’ Response

We have solved this problem. (Revised MS Lines 110)

Comment 11:
L 124. Change “S. thermalis” to “Schizopygopsis thermalis

Authors’ Response

We have solved this problem. (Revised MS Lines 124)

Comment 12

L 199. Change “morph” to “morphs”

Authors’ Response

We have solved this problem. (Revised MS Lines 200)

Comment 13:

L 199. Change “S. thermalis” to “Schizopygopsis thermalis

Authors’ Response

We have solved this problem. (Revised MS Lines 201)

Comment 14:

L 216. Change “according to the” to “according to”

Authors’ Response

We have solved this problem. (Revised MS Lines 217)

Comment 15:

L 216. Change “S. thermalis” to “Schizopygopsis thermalis

Authors’ Response

We have solved this problem. (Revised MS Lines 217)

Comment 16:

L 17. Change “they mean” to “the mean”

Authors’ Response

We have solved this problem. (Revised MS Lines 17)

Comment 17:

L 20. Change “morph” to “morphs”

Authors’ Response

We have solved this problem. (Revised MS Lines 20)

Comment 18:

L 20. Change “S. thermalis” to “Schizopygopsis thermalis

Authors’ Response

We have revised this problem. (Revised MS Lines 20)

Comment 19:

L 30. Change “The student’s t-test” to “Student’s t-test”

Authors’ Response

We have revised this problem. (Revised MS Lines 30)

Comment 20:

L 46. Change “student’s t test” to “Student’s t-test”

Authors’ Response

We have revised this problem. (Revised MS Lines 48)

Comment 21:

L 50. Change “student’s t test” to “Student’s t-test”

Authors’ Response

We have revised this problem. (Revised MS Lines 52)

Comment 22:

L 50. Change “S. thermalis” to “Schizopygopsis thermalis

Authors’ Response

We have revised this problem. (Revised MS Lines 52)

Comment 23:

L 72. Change “lakes” to “fishes”

Authors’ Response

We have changed “lakes” to “fishes”. (Revised MS Lines 61)

Comment 24:

L 73. Change “in fishing management” to “for fishing management”

Authors’ Response

We have changed “in fishing management” to “for fishing management”. (Revised MS Lines 62)

Comment 25:

L 134. Change “between two morphs” to “between the two morphs”

Authors’ Response

We have changed “between two morphs” to “between the two morphs”. (Revised MS Lines 123)

Reviewer 3 Report

The manuscript is very well written; clear, precise, and easy to understand.

The theme of the manuscript is interesting.

I have provided some remarks on the text and I  made additional suggestions (e.g. on statistical analysis).

Author Response

Reviewer(s)’ Comments to Author:

Reviewer# 3:

Authors’ Response

We are very grateful for the reviewer’s careful reading of our manuscript. We very much appreciate the suggestions made by the reviewer, and we have carefully addressed all the comments to improve our manuscript. Thank you very much for pointing out many details of the manuscript. We have modified the manuscript one by one. Please check the manuscript. For the other two specific questions, we will make specific replies below.

Core comment 1:

Line 167. In Group 1, the number of the individuals of the two morphs are 34 and 37 respectively. Since the Shapiro-Wilk test is more appropriate for small sample sizes (< 50 samples), it is preferable to use Shapiro-Wilk instead of Kolmogorov-Smirnov statistical analysis of normality.

Authors’ Response

We have replenished the method for normal analysis. Since our sample sizes in different groups are distinct, we add the Shapiro-Wilk test to the “2.4. Statistical analysis” to test for small sample sizes (< 50 samples), and the Kolmogorov-Smirnov test to test large sample sizes (> 50 samples). (Revised MS Lines 166-167) Correspondingly, we also re-tested the normality of the group 1 samples, and the result was similar to the original result.

Core comment 2:

Line 36. provide the results of the chi-square test (e.g. Χ2 (1, n=?) = result of chi square)

Authors’ Response

We have added the results of the chi-square test as follow “Because the chi-square test showed that the sex ratio was not significantly different between the two morphs in each TL group (all TL groups: χ2 (1, n = 432) = 0.285; first TL group: χ2 (1, n = 71) = 0.037; second TL group: χ2 (1, n = 234) = 3.243, and third TL group: χ2 (1, n = 127) = 0.001; all p > 0.05)”. (Revised MS Lines 36-38)
